# GEODESIC CALCULUS ON LATENT SPACES

## ABSTRACT

Latent manifolds of autoencoders provide low-dimensional representations of data, which can be studied from a geometric perspective. We propose to describe these latent manifolds as implicit submanifolds of some ambient latent space. Based on this, we develop tools for a discrete Riemannian calculus approximating classical geometric operators. These tools are robust against inaccuracies of the implicit representation often occurring in practical examples. To obtain a suitable implicit representation, we propose to learn an approximate projection onto the latent manifold by minimizing a denoising objective. This approach is independent of the underlying autoencoder and supports the use of different Riemannian geometries on the latent manifolds. The framework in particular enables the computation of geodesic paths connecting given end points and shooting geodesics via the Riemannian exponential maps on latent manifolds. We evaluate our approach on various autoencoders trained on synthetic and real data.

## 1 INTRODUCTION

In machine learning, extracting low-dimensional data representations is a classical problem, motivated by the manifold hypothesis that many high-dimensional datasets, such as images, lie on or near low-dimensional submanifolds. Approaches range from classical manifold learning methods, such as Isomap (Tenenbaum et al., 2000) and Diffusion Maps (Coifman et al., 2005), to neural network-based methods, including autoencoders, their probabilistic variants (e.g., variational autoencoders, VAE) (Kingma & Welling, 2013), and Generative Adversarial Networks (Goodfellow et al., 2014).

These ideas remain central in modern machine learning, for instance, in word embeddings for large language models (Devlin et al., 2019) or autoencoders in diffusion models (Rombach et al., 2022). Low-dimensional representations of the data manifold are crucial for high-performing generative models, yet their rich information (e.g., intrinsic dimension, topology, or point proximity) is rarely used explicitly. Instead, they are primarily considered an intermediate compression step.

In contrast, shape analysis extensively exploits manifold representations of geometric data. Riemannian manifolds—manifolds with a local measure of length—are a standard tool for modeling collections of shapes called shape spaces. Derived geometric operators are central for celebrated methods like LDDMM (Beg et al., 2005), enabling applied tasks to be phrased in terms of Riemannian calculus, e.g., shape interpolation via computing interpolating geodesics and shape extrapolation via the exponential map.

Yet, evaluating Riemannian operations on shape spaces is often computationally expensive, partly due to high dimensionality. Autoencoders could efficiently parametrize low-dimensional shape submanifolds, but their latent spaces typically lack explicit geometric structure. This highlights an open challenge in manifold learning: equipping latent spaces with geometric structure and practically usable geometric operators.

Previous work has focused on learning underlying structures, e.g., manifold representations (Arvanitidis et al., 2018) or Riemannian metrics (Gruffaz & Sassen, 2025). However, practical computation of geodesic interpolation between given endpoints or geodesic extrapolation remains challenging. We address this by making the latent manifold's geometry accessible via an implicit representation based on a learned projection that minimizes a denoising objective. We further introduce a time-discrete variational geodesic calculus suitable for imperfect implicit representations, along with practical computational algorithms. Figure 1 shows an illustrative example for both components. Building on time discretizations proven effective for Riemannian shape spaces (Rumpf & Wirth,

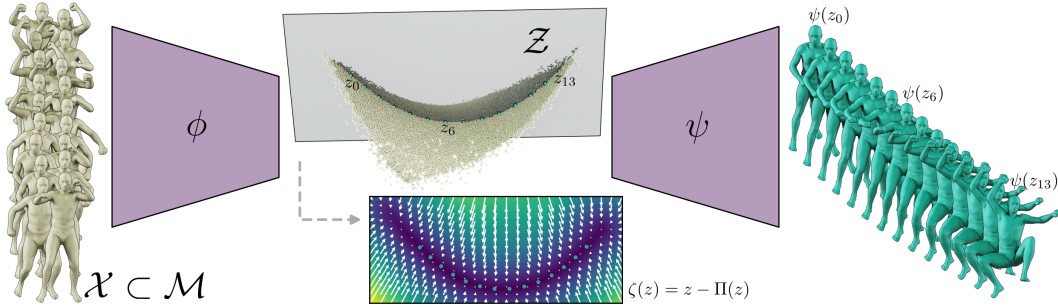

Figure 1: Training an autoencoder $(\phi, \psi)$ with data $\mathcal{X}$ lying on a manifold $\mathcal{M}$ yields a low-dimensional latent manifold $\mathcal{Z}$. Using a denoising objective, we learn an implicit representation $\zeta$ of this manifold (color-coding on the 2D slice from blue to yellow indicates $|\zeta|$) based on a projection $\Pi$ onto $\mathcal{Z}$ (white arrows, rescaled). Furthermore, we introduce a practical geodesic calculus on this representation, enabling, e.g., shape interpolation using latent manifolds (green dots and shapes).

2015), we hope to open up new ways of using latent representations in machine learning in general and enable new reduced-order methods for shape spaces in particular.

**Contributions.**   In summary, we make the following contributions:

○ We introduce a time-discrete geodesic calculus for imperfect representations of implicit latent manifolds and provide algorithms for geodesic interpolation and extrapolation.
○ We suggest minimizing a denoising objective to learn an approximate projection on latent manifolds with unknown codimension.
○ We evaluate our approach on various autoencoders trained on synthetic and real data.
○ We provide code in an easy-to-use fashion for different metrics and implicit representations. [Link will be provided in final version]

## 1.1   RELATED WORK

**Latent Space Geometry.**   Their widespread use has made the latent space of autoencoders a prime object of study, and equipping them with an appropriate notion of geometry is a major goal. Shao et al. (2018) compute interpolations and other geometric operations on the data manifold by parametrizing it via the decoder, which is equivalent to pulling back the Euclidean metric from the data space to the latent space. However, they disregard any non-Euclidean geometric structure of the latent space. Chen et al. (2018) pursue a similar idea for VAEs, where they additionally modify the pulled-back metric to generate high cost away from the data manifold. This underlying idea was concurrently pursued by Arvanitidis et al. (2018) based on previous work by Tosi et al. (2014) on Gaussian process latent variable models. They further extended this idea to pulling back non-Euclidean metrics (Arvanitidis et al., 2021), to pulling back the Fisher–Rao metric on densities (Arvanitidis et al., 2022; Lobashev et al., 2025), and to using Finsler metrics on latent spaces (Pouplin et al., 2023). In contrast to these works, we propose to encode the latent manifold as an implicit submanifold and derive a discrete geodesic calculus from this description. Sun et al. (2025) also learn an implicit representation of the latent manifold, however, they use it to modify the metric on the latent space and pull this modified metric back to the data space via the encoder. We will perform all optimizations directly on the latent manifold to maintain the advantages of its low dimensionality.

**Discrete Geodesic Calculus.**   One of the most fundamental tasks in Riemannian geometry is the computation of geodesic paths, either solving the system of geodesic differential equations via numerical integration techniques for given initial data, as in (Beg et al., 2005), or minimizing the so-called path energy over paths with prescribed endpoints. The latter variant is based on a suitable discretization of the path energy depending on the type of Riemannian space. In section 3 we will follow this time discretization paradigm. By (Rumpf & Wirth, 2015) it was developed into a com-

prehensive time-discrete geodesic calculus on shape spaces (including a corresponding convergence analysis for vanishing time steps), and it was applied in different contexts such as curves (Bauer et al., 2017), discrete surfaces (Heeren et al., 2014), or images (Berkels et al., 2015).

**Neural Implicits.** Implicit representations of geometric objects using neural networks have emerged as a new paradigm in computer graphics and computer vision, sparking broad research (Essakine et al., 2025). In particular, neural signed distance functions are often used to describe surfaces in three dimensions (Schirmer et al., 2024). Most work in this direction exclusively focuses on representing objects in three-dimensional space. However, we want to represent implicit submanifolds of arbitrary dimension and codimension. For this, signed distance functions are not a suitable implicit representation, and we will explore learning approximate projections instead in section 4.

## 2 BACKGROUND: RIEMANNIAN GEOMETRY

In this paper, we consider an autoencoder for data $\mathcal{X} \subset \mathcal{M} \subset \mathbb{R}^n$ on a hidden manifold $\mathcal{M}$ consisting of an *encoder* $\phi \colon \mathbb{R}^n \to \mathbb{R}^l$ and *decoder* $\psi \colon \mathbb{R}^l \to \mathbb{R}^n$. We will describe the corresponding latent manifold $\mathcal{Z} := \phi(\mathcal{M})$ as an implicit submanifold with a Riemannian metric and develop suitable numerical schemes for geodesic interpolation and extrapolation. Let us shortly recap the basics of the required Riemannian geometry: We consider the manifold $\mathcal{Z} \subset \mathbb{R}^l$ as an implicit, $m$-dimensional submanifold, i.e. we have $\mathcal{Z} := \left\{ z \in \mathbb{R}^l \mid \zeta(z) = 0 \right\}$ for a smooth map $\zeta$. In our case, $\zeta \colon \mathbb{R}^l \to \mathbb{R}^l$; $\zeta(z) := z - \Pi(z)$, where $\Pi \colon \mathbb{R}^l \to \mathbb{R}^l$ is the projection from the latent space $\mathbb{R}^l$ to the nearest point on the latent manifold $\mathcal{Z}$. The tangent space $T_z\mathcal{Z}$ at a point $z \in \mathcal{Z}$ is the vector space of all velocities of paths passing through $z$. For implicit submanifolds, $T_z\mathcal{Z} = \ker D\zeta(z)$. A *Riemannian metric* is an inner product $g_z(\cdot, \cdot)$ on $T_z\mathcal{Z}$ smoothly depending on $z$. This inner product allows to measure lengths of tangent vectors and angles between them. The simplest choice would be the Euclidean inner product inherited from $\mathbb{R}^l$, but it may be useful to apply other inner products that better represent the geometric structure of the original data.

The length of a path $\mathbf{z} \colon [0,1] \to \mathcal{Z}$ with velocity $\dot{\mathbf{z}}$ is defined as $\mathcal{L}(\mathbf{z}) = \int_0^1 \sqrt{g_{\mathbf{z}(t)}(\dot{\mathbf{z}}(t), \dot{\mathbf{z}}(t))}\ \mathrm{d}t$, and the *Riemannian distance* $\mathrm{dist}(z_0, z_1)$ between two points $z_0, z_1 \in \mathcal{Z}$ is the infimum over all paths with endpoints $\mathbf{z}(0) = z_0$, $\mathbf{z}(1) = z_1$. A minimizing path is called a *geodesic*. A torus $\mathcal{Z}$ is shown as a toy example for $l = 3$ in fig. 2. A geodesic connecting $z_0$ and $z_1$ can equivalently be found by minimizing the path energy

$$\mathcal{E}(\mathbf{z}) = \int_0^1 g_{\mathbf{z}(t)}(\dot{\mathbf{z}}(t), \dot{\mathbf{z}}(t))\ \mathrm{d}t \tag{1}$$

over curves $(\mathbf{z}(t))_{t \in [0,1]}$ in $\mathbb{R}^l$, subject to $\mathbf{z}(0) = z_0$, $\mathbf{z}(1) = z_1$ and $\zeta(\mathbf{z}) = 0$. Physically, geodesics have vanishing acceleration within the manifold, i.e. they always go straight at constant speed, neither changing direction nor velocity. Of course, viewed from the outside, a geodesic path on a curved manifold no longer looks straight. For the Euclidean inner product inherited from the ambient $\mathbb{R}^l$ as metric, the corresponding geodesic equation (the Euler–Lagrange equation associated with the minimization of $\mathcal{E}$) expresses the lack of acceleration *within* the manifold, $\ddot{\mathbf{z}}(t) \perp T_{\mathbf{z}(t)}\mathcal{Z}$. For implicit submanifolds, this reads as $D\zeta(\mathbf{z}(t))\dot{\mathbf{z}}(t) = 0$ and $\ddot{\mathbf{z}}(t) \cdot w = 0$ for all $D\zeta(\mathbf{z}(t))w = 0$. For other Riemannian metrics, this geodesic ODE defining a geodesic for initial data $\mathbf{z}(0) = z$ and $\dot{\mathbf{z}}(0) = v \in T_z\mathcal{Z}$ becomes more complicated. Mapping $v$ to the arrival point $y = \mathbf{z}(1)$ at time 1 yields the *Riemannian exponential map* $\exp_z \colon T_z\mathcal{Z} \to \mathcal{Z}\ \colon\ \exp_z v = y$.

## 3 DISCRETE GEODESIC CALCULUS

We introduce a time-discrete geodesic calculus suitable for (imperfect) implicit manifold representations. A central ingredient is a local approximation $\mathcal{W}(\cdot, \cdot)$ of the squared Riemannian distance used to define the *discrete path energy*

$$\mathcal{E}^K(z_0, \ldots, z_K) = K \sum_{k=1,\ldots,K} \mathcal{W}(z_{k-1}, z_k) \tag{2}$$

of a discrete path $\mathbf{z} = (z_0, \ldots, z_K)$ as a discrete counterpart of (1). Consequently, a *discrete geodesic* for given endpoints $z_0, z_K \in \mathcal{Z}$ is a minimizer of (2) subject to the constraint $\zeta(z_k) = 0$

for $k = 0, \ldots, K$. Rumpf & Wirth (2015, Corollary 4.10) have shown that if $\mathcal{Z}$ is smooth, $z \mapsto g_z$ is Lipschitz continuous, and $\mathcal{W}(z_0, z_1)$ approximates $\mathrm{dist}(z_0, z_1)^2$ up to an error $O(\mathrm{dist}(z_0, z_1)^{-3})$, then the piecewise affine interpolations of minimizers of the discrete path energies $\mathcal{E}^K$ converge uniformly to minimizers of the continuous path energy $\mathcal{E}$. One can interpret $\mathcal{W}(z_{k-1}, z_k)$ physically as the energy of a spring connecting $z_{k-1}$ and $z_k$. Then, $\mathcal{E}^K(z_0, \ldots, z_K)$ is the total elastic energy of a chain of $K$ springs, which we relax under the constraint that all nodes lie on $\mathcal{Z}$. Depending on the application and the underlying configuration of the data manifold, one can distinguish different Riemannian metrics on $\mathcal{Z}$ and corresponding functionals $\mathcal{W}$, for instance,

- the Euclidean inner product $g_z(v, w) = v \cdot w$ inherited from the ambient space $\mathbb{R}^l$ leads to

$$\mathcal{W}_{\mathrm{E}}(z_0, z_1) \coloneqq |z_1 - z_0|^2, \tag{3}$$

- the *pullback metric* $g_z(v, w) = g_{\psi(z)}^{\mathcal{M}}(D\psi(z)v, D\psi(z)w)$ pulling back the metric $g_x^{\mathcal{M}} : T_x\mathcal{M} \times T_x\mathcal{M} \to \mathbb{R}$ of $\mathcal{M}$ results in

$$\mathcal{W}_{\mathcal{M}}(z, \tilde{z}) = \mathrm{dist}_{\mathcal{M}}^2(\psi(z), \psi(\tilde{z})), \tag{4}$$

- if $\mathcal{M}$ is equipped with the Euclidean inner product inherited from $\mathbb{R}^n$, $\mathcal{W}_{\mathcal{M}}$ simplifies to

$$\mathcal{W}_{\mathrm{PB}}(z, \tilde{z}) = |\psi(z) - \psi(\tilde{z})|^2. \tag{5}$$

We discuss this further in appendix A.1.

**Computing discrete geodesics.** The Lagrangian for the constrained optimization problem of minimizing (2) subject to $\zeta(\mathbf{z}) = 0$ reads $\mathbf{L}(\mathbf{z}, \Lambda) = \mathcal{E}^K(\mathbf{z}) - \Lambda : \zeta(\mathbf{z})$, where $\mathbf{z} \in \mathbb{R}^{l, K+1}$ and $\Lambda \in \mathbb{R}^{l, K-1}$ denotes the matrix of the components of the Lagrange multiplier and $\Lambda : \zeta(\mathbf{z}) \coloneqq \sum_{i=1}^l \sum_{k=1}^{K-1} \Lambda_{ik}\zeta_i(z_k)$. The optimality conditions for the constrained optimization can be expressed as the saddle point condition

$$0 = \partial_{z_k}\mathbf{L}(\mathbf{z}, \Lambda) = K\left(\partial_{z_k}\mathcal{W}(z_{k-1}, z_k) + \partial_{z_k}\mathcal{W}(z_k, z_{k+1})\right) - \sum_{i=1,\ldots,l} \Lambda_{ik}\nabla\zeta_i(z_k), \tag{6}$$

$$0 = \partial_{\Lambda_{ik}}\mathbf{L}(\mathbf{z}, \Lambda) = \zeta_i(z_k) \tag{7}$$

for $k = 1, \ldots, K-1$ and $i = 1, \ldots, l$.

We propose to use an augmented Lagrangian method to compute solutions of the constrained optimization problem. In detail, for the augmented Lagrangian

$$\mathbf{L}^a(\mathbf{z}_j, \Lambda_j, \mu_j) = \mathbf{L}(\mathbf{z}_j, \Lambda_j) + \tfrac{\mu_j}{2}|\zeta(\mathbf{z}_j)|^2 = \mathcal{E}(\mathbf{z}_j) - \Lambda_j : \zeta(\mathbf{z}_j) + \tfrac{\mu_j}{2}|\zeta(\mathbf{z}_j)|^2 \tag{8}$$

we iterate the update rules

$$\mathbf{z}_{j+1} = \operatorname*{arg\,min}_{\mathbf{z} \in \mathbb{R}^{l(K-1)}} \mathbf{L}^a(\mathbf{z}, \Lambda_j, \mu_j), \tag{9}$$

$$\Lambda_{j+1} = \Lambda_j - \mu_j\zeta(\mathbf{z}_{j+1}), \quad \mu^{j+1} = \alpha\,\mu^j \tag{10}$$

for given initial data $(\mathbf{z}_0, \Lambda_0, \mu_0)$ and some $\alpha > 1$. The update of the multiplier $\Lambda$ ensures that the Euler–Lagrange equation $\partial_{\mathbf{z}}\mathbf{L}^a = 0$ coincides with the first saddle point condition (6). The third term of $\mathbf{L}^a$ is a penalty ensuring closeness of $\mathbf{z}_j$ to $\mathcal{Z}$ and thus reflects the second saddle point condition (7).

The augmented Lagrangian approach is not harmed by the fact that the Lagrange multiplier in (6)-(7) is underdetermined (and thus nonunique) due to rank $D\zeta(z) = l - m$. Moreover, it is also applicable with inexact constraints (Frick et al., 2011; Jin, 2017), which is important since in practice we replace $\zeta$ with $\zeta_\sigma = \mathrm{id} - \Pi_\sigma$ for a learned *approximate* projection $\Pi_\sigma$ (cf. section 4). The augmented Lagrangian method allows this approximation as long as $\zeta_\sigma$ points approximately in normal direction to $\mathcal{Z}$. Furthermore, the penalty term enforces small values of $\zeta_\sigma(z_k) = z_k - \Pi_\sigma(z_k)$ even though $\zeta_\sigma$ is not expected to vanish. Overall, we observe that the augmented Lagrangian method works for an inexact implicit function $\zeta_\sigma$ as long as $(\zeta_\sigma, D\zeta_\sigma)$ approximate $(\zeta, D\zeta)$ sufficiently well (see, e.g., fig. 2). We give details on the implementation and the parameters for the augmented Lagrangian approach in appendix A.2.

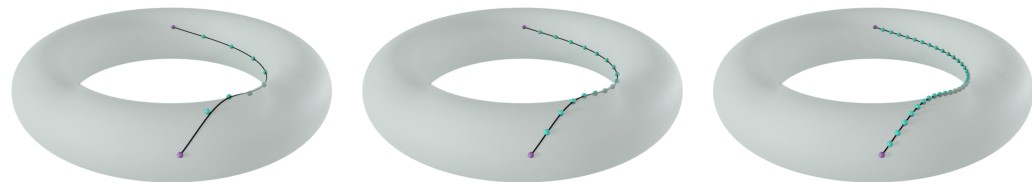

Figure 2: Discrete geodesics for different values of $K$ computed on a torus with learned implicit manifold representation $\zeta_\sigma$ (green points) and highly resolved geodesic computed with ground truth representation $\zeta$ (black line).

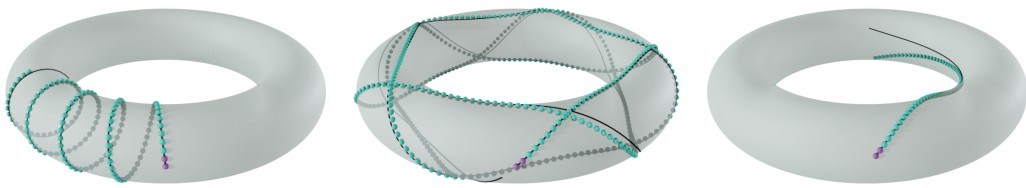

Figure 3: Comparison between computed exponentials with learned implicit manifold representation $\zeta_\sigma$ (green points) and ground truth representation $\zeta$ (black line). As in any dynamical system, slight numerical inaccuracies lead to an exponentially growing divergence (which is known to be more pronounced in regions of negative curvature as in the right-most example).

**Computing a discrete exponential.** To derive a discrete counterpart of geodesic extrapolation via the exponential map, we interpret the first saddle point condition

$$0 = K(\partial_{z_k}\mathcal{W}(z_{k-1}, z_k) + \partial_{z_k}\mathcal{W}(z_k, z_{k+1})) - \nabla\zeta(z_k)\lambda_k$$

from the discrete geodesic interpolation (cf. (6)) as a (nonlinear) equation in the unknowns $z_{k+1}$ and $\lambda_k$ for given $z_{k-1}$, $z_k$. Furthermore, we implement the constraint $\zeta(z_k) = 0$ from the second saddle point condition (7) as a penalty. This, leads to the minimization of the functional $\mathcal{F} : \mathbb{R}^l \times \mathbb{R}^l \to \mathbb{R}$,

$$\mathcal{F}(z_{k+1}, \lambda_k) := |K(\partial_{z_k}\mathcal{W}(z_{k-1}, z_k) + \partial_{z_k}\mathcal{W}(z_k, z_{k+1})) - \nabla\zeta(z_k)\lambda_k|^2 + \tfrac{\mu}{2}|\zeta(z_{k+1})|^2, \quad (11)$$

where $\mu > 0$ is some penalty parameter. As in the context of the discrete geodesic interpolation, this minimization remains reasonable if $\zeta$ is replaced by an approximation $\zeta_\sigma$. Now, given the first two points $z_0$ and $z_1$ and thus a corresponding discrete initial velocity $v_0 = K(z_1 - z_0)$, we iteratively minimize $\mathcal{F}(z_{k+1}, \lambda_k)$ for $k = 1, \ldots, K - 1$ in both $z_{k+1}$ and $\lambda_k$ with a BFGS method. We define by $\text{Exp}_{z_0}^K(v_0) := z_K$ the *discrete exponential map* as the final point of the extrapolated discrete geodesic $(z_0, \ldots, z_K)$ (cf. fig. 3). Following Rumpf & Wirth (2015, Theorem 5.1), the discrete exponential map converges to the continuous one as $K \to \infty$.

## 4 PROJECTION AS IMPLICIT REPRESENTATION OF LATENT MANIFOLDS

In section 2, we defined the implicit function $\zeta(z) = z - \Pi(z)$ based on a projection $\Pi$ from the latent space to the latent manifold. In section 3, we considered a yet unspecified approximation $\zeta_\sigma(z) = z - \Pi_\sigma(z)$. In this section, we will detail the concrete choice of $\zeta_\sigma$ or rather of the approximate projection $\Pi_\sigma$ as the minimizer of a suitable objective and discuss how to train its network representation.

**Projection as minimizer of a loss functional.** Let us suppose a density measure $\text{d}z$ is given on the latent manifold $\mathcal{Z}$ reflecting the sampling on the data manifold. In applications, this is typically the empirical measure of the data samples or rather of their images under encoder $\phi$. The projection as a neural network is learned based on this possibly noisy point cloud representing the latent manifold.

Following the approach to learn a projection from denoising autoencoders proposed by Alain & Bengio (2014), we define the projection $\Pi_\sigma$ as the minimizer of the loss functional

$$\mathcal{Q}(\Pi) = \int_{\mathbb{R}^l} \int_{\mathcal{Z}} |z - \Pi(y)|^2 f_\sigma(z - y) \, \text{d}z \, \text{d}y \qquad (12)$$

over maps $\Pi : \mathbb{R}^l \to \mathbb{R}^l$. Here, $f_\sigma(y)$ is the normal distribution with mean 0 and standard deviation $\sigma$. The functional $\mathcal{Q}$ is a coercive quadratic form. Hence, the condition that $\partial_\Pi \mathcal{Q}(\Pi_\sigma)$ vanishes uniquely classifies the minimizer $\Pi_\sigma$ and leads to $0 = 2 \int_{\mathbb{R}^l} \int_{\mathcal{Z}} (\Pi_\sigma(y) - z) \vartheta(y) f_\sigma(z - y) \, dz \, dy$ for all $\vartheta \in C_c^\infty$. Thus, one obtains the approximate projection

$$\Pi_\sigma(y) = \left( \int_{\mathcal{Z}} f_\sigma(y - z) \, dz \right)^{-1} \int_{\mathcal{Z}} z f_\sigma(y - z) \, dz$$

of $y$ in the neighborhood of $\mathcal{Z}$ as a Gaussian-weighted $\mathcal{Z}$–barycenter (cf. Alain & Bengio (2014)). For $\mathcal{Z}$ being an affine subspace of $\mathbb{R}^l$, $\Pi_\sigma$ is indeed the orthogonal projection on $\mathcal{Z}$. In general, $\mathrm{id} - \Pi_\sigma$ does not necessarily vanish on $\mathcal{Z}$ but comes with a defect of order $O(\sigma^2)$ in the relative interior of smooth latent manifolds and $O(\sigma)$ close to the boundary.

**Learning the projection on encoded samples.** To practically minimize the objective (12), we parameterize $\Pi_\sigma$ by a fully connected neural network with ELU activation functions (Clevert et al., 2015). We minimize the objective (12) using the Adam optimizer (Kingma & Ba, 2014).

In appendix A.3, we study the properties of the denoising loss and the resulting $\Pi_\sigma$ on a low-dimensional toy model and show experimentally that the approximation error decreases for increasing point cloud size, increasing network architectures, and decreasing noise levels.

In applications, we have an approximation of the latent manifold $\mathcal{Z}$ by a point cloud $\phi(\mathcal{X})$ of encoded data samples $\mathcal{X}$. These point clouds can be sparse and noisy depending on the distribution of the data and the regularization of the autoencoder. Suitable values of $\sigma$ depend on the sampling. For data with low noise level and high point cloud density, small values of $\sigma$ are preferable as long as the convolution with the Gaussian $f_\sigma$ sufficiently regularizes the data distribution. On the other hand, a reliable projection further away from $\mathcal{Z}$ can only be expected for sufficiently large $\sigma$. In our experiments below, we could choose the same $\sigma$ for similar point cloud densities.

## 5 RESULTS

In the following examples, we demonstrate the performance of the method across different types of data and latent manifolds obtained from different autoencoders.

### 5.1 DISCRETE SHELLS / ISOMETRIC AUTOENCODER

We compute interpolations and extrapolations on a submanifold of the space of discrete shells (Grinspun et al., 2003), i.e., the space of all possible immersions of a fixed triangle mesh, and equip this space with the Riemannian metric proposed by Heeren et al. (2014). First, we train an isometric autoencoder to approximate this submanifold and then the projection operator as described in section 4, allowing us to perform the geodesic calculus on the latent manifold.

The submanifold is designed to approximate a dataset of shapes, such as different poses of a humanoid model. Because many datasets provide only a limited number of examples, we first apply a classical Riemannian construction to obtain the submanifold and its parametrization, which is computationally demanding. We then use this parametrization to create a denser training set for our autoencoder. We discard any pairs where at least one shape exhibits self-intersections to preserve physical plausibility. See appendix A.4, for details on the data generation and the autoencoder training. In this way, the autoencoder learns a low-dimensional latent manifold approximating the shape space submanifold, enabling us to perform discrete geodesic operations in reduced dimensions. Figure 1 illustrates the overall procedure of our approach for an example of a two-dimensional submanifold of the manifold of discrete shells.

**Geodesic calculus on the latent manifold.** As described in section 4, we use the trained projection operator $\Pi_\sigma$ to construct an approximate implicit representation of the latent manifold. For the geodesic calculus, we use the Euclidean metric, as this agrees with the shell distance due to the autoencoder's isometry. In fig. 4, we show a result of applying this approach to the SCAPE dataset (Anguelov et al., 2005) of human character poses and compare linear interpolation in latent space with geodesics on the latent manifold computed using our approach. Our geodesics avoid self-intersections more effectively than linear interpolation, thanks to the rejection of intersecting shapes

Figure 4: Interpolations on a learned submanifold of the shape space of discrete shells. Comparison between linear interpolation in latent space (red) and geodesic interpolation using a learned implicit representation (green) of the latent manifold $\mathcal{Z}$.

during training. Since our learned projection $\Pi_\sigma$ is only accurate up to the filter width $\sigma$, slight self-intersections may remain. We show additional examples in appendix A.4.

## 5.2 Motion capture data / spherical variational autoencoder

We consider a latent manifold resulting from training a spherical variational autoencoder (SVAE) Davidson et al. (2018) on motion capture data from the CMU Graphics Lab. A suitable representation for the data was described by Tournier et al. (2009): A pose is defined as an element of $\mathrm{SO}(3)^m$, where $m = 30$ is the number of joints in the skeleton. The vector of rotations specifies the rotations of the joints. Given the connectivity and lengths of the skeletal segments, the full pose can be reconstructed. Hence, our input data $\mathcal{X}$ lies on a hidden manifold $\mathcal{M} \subset \mathrm{SO}(3)^m$. Arvanitidis et al. (2022) used an SVAE autoencoder to take the hyperspherical nature of this data into account. Unlike standard VAEs, which assume a Gaussian prior in the latent space, the SVAE uses von Mises–Fisher (vMF) distributions for regularization, which indeed enforces a hyperspherical latent geometry. Details on the employed data and training are provided in appendix A.5. We use $l = 10$ latent dimensions. In the case of (S)VAEs, the encoder and decoder maps are not deterministic but parameterize distributions. We sample the latent manifold $\mathcal{Z}$ by sampling from the encoder distribution and obtain decoded points by sampling from the decoder. For simplicity, when learning $\Pi_\sigma$ and calculating discrete geodesics on $\mathcal{Z}$, we ignore the nondeterministic nature of the encoder and decoder and, by a slight abuse of notation, denote by $\psi(z) \in \mathrm{SO}(3)^m$ a sample of the decoding of $z$. Another approach would be to follow Arvanitidis et al. (2022) and incorporate the Kullback–Leibler divergence, see (14) in appendix A.1.

In figs. 5 and 6 (left), we show a projection of the sampled latent manifold (from which we learn $\Pi_\sigma$) onto the three most relevant dimensions obtained from a PCA.

**Geodesic calculus on the latent manifold.** The encoder embedding is not close to isometric. Hence, it is appropriate to equip the latent manifold with the pulled-back spherical distance

$$\mathcal{W}_{\mathcal{M}}(z, \tilde{z}) = \mathrm{dist}^2_{\mathrm{SO}(3)}(\psi(z), \psi(\tilde{z})) = \sum_{i=1}^{m} |\arccos(\psi(z)_i \cdot \psi(\tilde{z})_i)|^2. \tag{13}$$

In fig. 5 (right), we compare geodesic interpolation based on $\mathcal{W}_{\mathcal{M}}$ (green) to geodesic interpolation with the Euclidean metric $\mathcal{W}_{\mathrm{E}}(z, \tilde{z}) = |z - \tilde{z}|^2$ (yellow) as well as linear interpolation in latent space (red). The results show that geodesic interpolation with the pullback metric yields realistic decoded paths, whereas linear interpolation in latent space leads to poses not lying on the data manifold $\mathcal{M}$, as indicated by unnatural contractions of shoulders and hips and by self-intersecting limbs. Figure 6 shows a pose extrapolation using the exponential map on the latent manifold. Different from linear extrapolation, the extrapolated path follows the geometry of the latent manifold and, after decoding, leads to realistic poses.

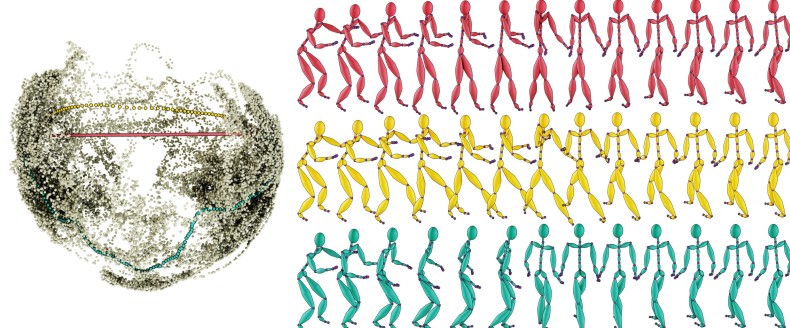

Figure 5: Left: Visualization of sample points in latent space (projected from $\mathbb{R}^{10}$ into $\mathbb{R}^3$ based on a PCA) and linear interpolation (red), geodesic interpolation with $\mathcal{W}_{\mathrm{E}}$ (yellow), and geodesic interpolation with $\mathcal{W}_{\mathcal{M}}$ (green). Right: Corresponding decoded sequences.

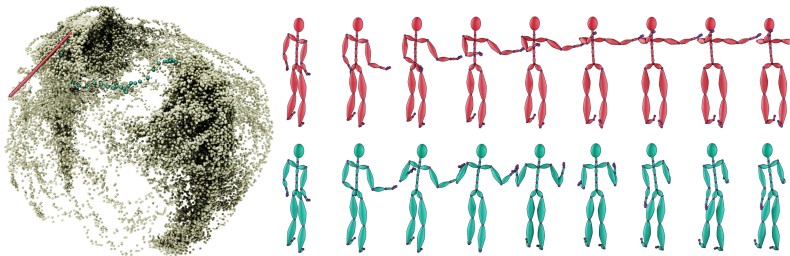

Figure 6: Left: Visualization of sample points in latent space (projected from $\mathbb{R}^{10}$ into $\mathbb{R}^3$ based on a PCA) and, starting from a fixed point in a fixed direction, linear extrapolation (red) and geodesic extrapolation with $\mathcal{W}_{\mathcal{M}}$ (green). Right: Corresponding decoded sequences.

### 5.3 IMAGE DATA / LOW BENDING, LOW DISTORTION AUTOENCODER

Next, we consider image data of a rotating three-dimensional object as proposed as an example by Braunsmann et al. (2024). We use their regularized autoencoder, minimizing a loss function that promotes the embedding to be as isometric and as flat as possible. The data $\mathcal{X} \subset \mathcal{M} \subset \mathbb{R}^{128 \times 128 \times 3}$ consists of RGB images showing a toy cow model (Crane et al., 2013) from varying viewpoints. Training the regularized autoencoder requires computing distances and averages between dataset samples. Each image $x$ corresponds to a specific rotation $r_x \in \mathrm{SO}(3)$ which allows to define these distances and averages. We use the publicly available pretrained autoencoder with $l = 16$-dimensional latent space, for details on other parameters and settings see appendix A.6. A PCA projection on three dimensions of the resulting latent manifold is visualized in fig. 7 (left), where the PCA analysis shows that the encoder uses six dimensions for the embedding.

**Geodesic calculus on the latent manifold.** As the encoder is regularized to be near-isometric, we use $\mathcal{W}_{\mathrm{E}}(z, \tilde{z}) = |z - \tilde{z}|^2$ in our geodesic calculus. In fig. 7, linear and geodesic interpolation as well as extrapolation in latent space are shown.

### 5.4 EXTENSION TO DISTANCE FUNCTION AS IMPLICIT REPRESENTATION

Finally, our discrete geodesic interpolation can be performed even if the latent manifold $\mathcal{Z}$ is merely represented by a distance function $d \colon \mathbb{R}^l \to \mathbb{R}$ with $\mathcal{Z} = \{z \in \mathbb{R}^l \mid d(z) = 0\}$. However, $d$ is non-differentiable on $\mathcal{Z}$. Hence, instead of the augmented Lagrange algorithm (9)-(10) we use the classical penalty method to minimize the discrete path energy $\mathcal{E}^K(\mathbf{z})$ subject to the constraints $d(z_k) = 0$. Computing the exponential map is not possible with missing normal information, though.

For example, Pose-NDF (Tiwari et al., 2022) provides a neural distance function to a manifold of plausible human poses. They use a quaternion representation of the SMPL body model by Loper et al. (2015), resulting in $l = 84$ dimensions. Comparisons with linear interpolation in these coordi-

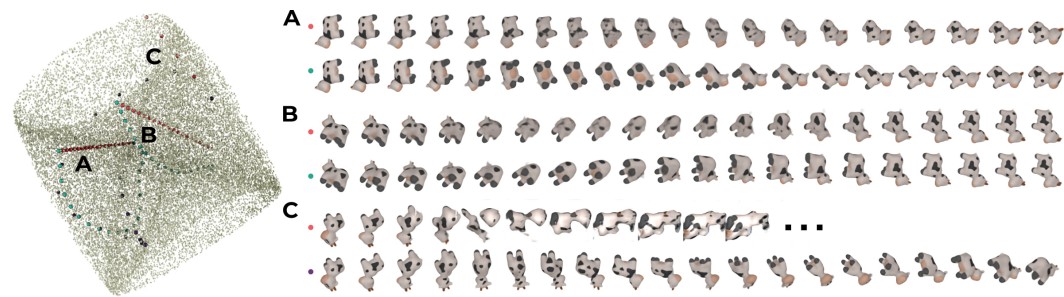

Figure 7: Left: Sample points in latent space (projected into $\mathbb{R}^3$; they represent an immersion of SO(3) which is topologically equivalent to the Klein bottle and thus has to self-intersect in three dimensions) and computed linear (red) and geodesic (green) interpolation (A, B) as well as linear (red) and geodesic (purple) extrapolation (C). Right: Corresponding decoded sequences.

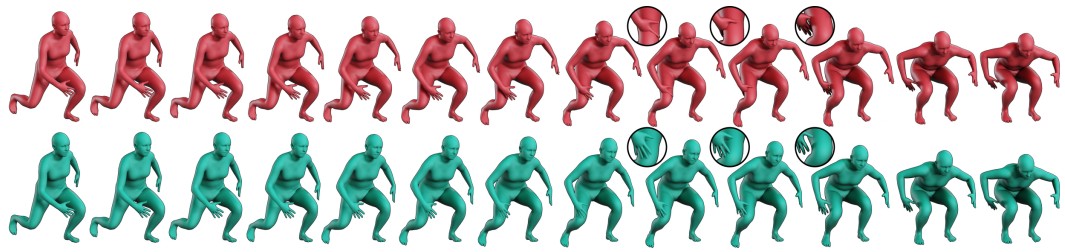

Figure 8: Linear interpolation in a quaternion representation of the SMPL body model (red) and geodesic interpolation on the manifold of plausible poses using the learned distance function provided by Tiwari et al. (2022) (green).

nates show that our approach computes geodesics on the manifold of plausible poses, whereas linear interpolation produces self-intersecting poses that lie off the manifold, see fig. 8. An additional example is provided in appendix A.7.

## 6 CONCLUSION

Our results demonstrate that geometric operations on a latent manifold $\mathcal{Z}$ are indeed feasible. Central to our approach is a learned projection $\Pi_\sigma$ onto $\mathcal{Z}$. Achieving efficiency, however, requires either fast distance evaluation in the data manifold or an embedding of $\mathcal{Z}$ that is (near-)isometric.

Several directions emerge for future work. A natural step is to connect the projection-based representation via $\Pi_\sigma$ with distance-based representations $d$, for instance by deriving $\Pi_\sigma$ directly from $d$. From a numerical perspective, key open questions include how the accuracy of $\Pi_\sigma$ (and of the resulting geometric operators) depends on the sampling density and the scale parameter $\sigma$, how to enhance imperfect projections (e.g., using multiple or adaptive $\sigma$ or exploiting the property $\Pi_\sigma \approx \Pi_\sigma \circ \Pi_\sigma$), and how to improve augmented Lagrangian techniques for inexact constraints.

Conceptually, the next step lies in moving from latent manifolds to latent distributions, particularly in the context of VAEs and related generative models. Along these lines, an interesting possibility is to replace our projection operator with conditional denoisers, as used in diffusion models. Finally, extending the calculus to support detail transfer via discrete parallel transport and curvature approximation would open further applications.

### ACKNOWLEDGMENTS

We used LLMs for minor language editing (e.g., grammar and style), all technical content was written by the authors.

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

## A  APPENDIX

### A.1  METRIC AND DISTANCE APPROXIMATION

We further discuss choices of the Riemannian metric and the corresponding functionals $\mathcal{W}$ to be used in our discrete geodesic calculus framework described in section 3.

○ The simplest choice for a metric on the latent manifold $\mathcal{Z}$ (though not particularly suitable for many applications) is the Euclidean inner product $g_z(v, w) = v \cdot w$ inherited from the ambient space $\mathbb{R}^l$. In this case,

$$\mathcal{W}_{\mathrm{E}}(z_0, z_1) := |z_1 - z_0|^2$$

is the obvious choice fulfilling the requirements on $\mathcal{W}$. Physically, this choice corresponds to the energy of a single Hookean spring.

○ If the data manifold $\mathcal{M}$ is embedded in $\mathbb{R}^n$ and equipped with the inherited Euclidean inner product, tangent vectors $v \in T_z\mathcal{Z} \subset \mathbb{R}^l$ correspond by the chain rule to embedded tangent vectors $D\psi(z)v \in \mathbb{R}^n$ after decoding. Hence the *pullback metric* (which gives tangent vectors to $\mathcal{Z}$ the same length as their counterparts on $\mathcal{M}$) is given by $g_z(v, w) = D\psi(z)v \cdot D\psi(z)w$. In this case, the squared Euclidean distance between the decoded points

$$\mathcal{W}_{\mathrm{PB}}(z, \tilde{z}) = |\psi(z) - \psi(\tilde{z})|^2$$

is an admissible approximation $\mathcal{W}$ of the squared Riemannian distance.

○ Frequently, the data manifold $\mathcal{M}$ is equipped with a metric $g_x^{\mathcal{M}} : T_x\mathcal{M} \times T_x\mathcal{M} \to \mathbb{R}$. Again, pulling this metric back to $\mathcal{Z}$ yields

$$g_z(v, w) = g_{\psi(z)}^{\mathcal{M}}(D\psi(z)v, D\psi(z)w) = D\psi(z)^T G_{\psi(z)}^{\mathcal{M}} D\psi(z)v \cdot w,$$

where $G_x^{\mathcal{M}}$ is the matrix representation of the metric $g_x^{\mathcal{M}}$. In applications where the Riemannian distance on $(\mathcal{M}, g^{\mathcal{M}})$ can be explicitly computed, one is naturally led to

$$\mathcal{W}_{\mathcal{M}}(z, \tilde{z}) = \mathrm{dist}_{\mathcal{M}}^2(\psi(z), \psi(\tilde{z}))$$

as a proper choice for $\mathcal{W}$, measuring the squared Riemannian distance of decoded points $\psi(z)$ and $\psi(\tilde{z})$.

○ In the case of non-deterministic decoders, $\psi(z)$ lies in a space of distributions. One can pull back the Fisher–Rao metric from the space of decoder distributions on the latent manifold. The Kullback–Leibler (KL)-divergence can then be used as a second-order distance approximation

$$\mathcal{W}_{\mathrm{KL}}(z, \tilde{z}) = \mathrm{KL}(\psi(z), \psi(\tilde{z})). \tag{14}$$

For a Gaussian decoder with fixed variances, where $\psi(z) = \mathcal{N}(\mu(z), \mathrm{I})$ is a normal distribution with mean $\mu(z)$, the KL-divergence reduces to the squared Euclidean distance of the means,

$$\mathrm{KL}(\psi(z), \psi(\tilde{z})) = \tfrac{1}{2}|\mu(z) - \mu(\tilde{z})|^2.$$

We refer to Arvanitidis et al. (2022) for details on this information geometry perspective.

### A.2  DETAILS ON THE AUGMENTED LAGRANGIAN METHOD

In practice, we use a slightly more advanced version of the Augmented Lagrangian method following the algorithm described by Nocedal & Wright (2006, Chapter 17), which we adapt to our setting in algorithm 1. Compared to the simplified version in the main text, we do not update the Lagrange multiplier and the penalty parameter in every iteration, but only depending on how well the constraint is already fulfilled. For the inner optimization problem, we use the BFGS method from SciPy (Virtanen et al., 2020). As the initial path we choose the path $\mathbf{z}_0 = (z_0 = z_0, \ldots, z_{\lfloor K/2 \rfloor} = z_0, z_{\lfloor K/2 \rfloor + 1} = z_K, \ldots, z_K = z_K)$ that remains constant and jumps directly from the given starting point $z_0$ to the endpoint $z_K$ at the middle time point. We set the initial Lagrange multiplier $\Lambda_{i0} = 0$ for $i = 1, \ldots, K$ and $\alpha = 2$. The final tolerance $\eta^*$ for the constraint is problem-dependent and depends on the minimum values of $\zeta_\sigma$. In practice, a good rule-of-thumb is to choose $\eta^*$ as $K$ times the mean value on the embedded data samples $\eta^* \approx \frac{K}{|\mathcal{X}|}|\zeta_\sigma(\phi(\mathcal{X}))|$.

**Algorithm 1** Augmented Lagrangian Method (Nocedal & Wright, 2006, Algorithm 17.4)

1: Choose initial point $\mathbf{z}_0$, multiplier $\Lambda_0$, penalty $\mu_0$, and $\alpha$
2: Choose final tolerances $\eta^*$ for constraint, $\omega^*$ for gradient, and maximum penalty $\mu_{\max}$
3: Set $\omega^0 \leftarrow 1/\mu^0$, $\eta^0 \leftarrow 1/\mu_0^{0.1}$
4: **for** $j = 0, 1, 2, \ldots$ **do**
5:     find approximate solution $\mathbf{z}_{j+1}$ of

$$\underset{\mathbf{z}\in\mathbb{R}^{l(K-1)}}{\arg\min} \ \mathbf{L}^a(\mathbf{z}, \Lambda_j, \mu_j)$$

6:     such that $|\nabla_{\mathbf{z}_{j+1}}\mathbf{L}^a(\mathbf{z}_{j+1}, \Lambda_j, \mu_j)| \le \omega^k$
7:     **if** $|\zeta(\mathbf{z}_{j+1})| \le \eta^k$ **then**
8:         **if** $|\zeta(\mathbf{z}_{j+1})| \le \eta^*$ **and** $|\nabla_{\mathbf{z}_{j+1}}\mathbf{L}^a(\mathbf{z}_{j+1}, \Lambda_j, \mu_j)| \le \omega^*$ **then**
9:             **return** $\mathbf{z}_{j+1}$                    ▷ final accuracy reached
10:         **end if**
11:         update multiplier

$$\Lambda_{j+1} = \Lambda_j - \mu_j\zeta(\mathbf{z})$$

12:         update tolerances

$$\mu_{j+1} = \mu_j, \quad \eta_{j+1} = \eta_j/\mu_{j+1}^{0.9}, \quad \omega_{j+1} = \omega_j/\mu_{j+1}$$

13:     **else**
14:         increase penalty parameter

$$\mu_{j+1} \leftarrow \alpha\mu_j$$

15:         update tolerances

$$\Lambda_{j+1} = \Lambda_j, \quad \eta_{j+1} \leftarrow 1/\mu_{j+1}^{0.1}, \quad \omega_{j+1} \leftarrow 1/\mu_{j+1}$$

16:
17:         **if** $\mu_{j+1} > \mu_{\max}$ **then**
18:             **return** $\mathbf{z}_{j+1}$                    ▷ max penalty reached
19:         **end if**
20:     **end if**
21: **end for**

### A.3 Details: Learning the projection on encoded samples.

We provide additional details on the learned projection (section 4) used to construct an implicit representation of the latent manifolds.

**Optimization parameters.**   We train a fully connected neural network with ELU activations (Clevert et al., 2015) by minimizing the loss functional (12) using the Adam optimizer (Kingma & Ba, 2014) with a learning rate of $10^{-3}$ and a weight decay of $10^{-5}$. The layer dimensions depend on the examples. The batch size to evaluate the integral over $\mathcal{Z}$ is 128 in all our examples. To approximate the inner integral, we sample a single point $y_z$ from $f_\sigma$ for each data sample $z$ and optimization step.

**Parameter study.**   To analyze the denoising loss and the resulting approximate projection $\Pi_\sigma$ for different parameters, we use the toy torus model to allow comparison with a ground truth projection and keep the evaluation visually tractable. In practice, only approximations of the latent manifold $\mathcal{Z}$ are available. To study the denoising property, we generate a noisy torus surface and train a projection with different values for $\sigma$, treating the noisy surface as $\mathcal{Z}$. We then visualize the image $\Pi_\sigma(\mathcal{Z})$ under the projections. As expected, a larger value of $\sigma$ leads to a stronger smoothing effect, see fig. 9.

For a point cloud without noise, a small parameter $\sigma$ leads to a higher accuracy of $\Pi_\sigma$ close to the surface but larger errors at certain distances, as those points are rarely seen in training, see fig. 10 (left). We further evaluate in fig. 10 the stability of the optimization, showing that the approximation error decreases for increasing point cloud size, increasing network architectures, and decreasing noise levels.

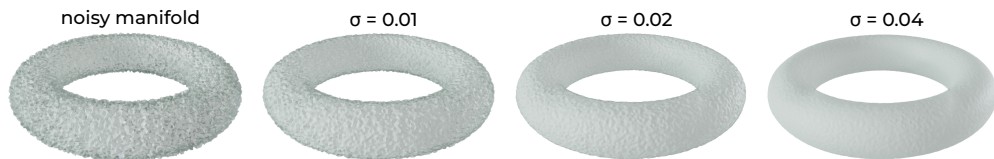

Figure 9: Visualization of the denoising effect for different choices of $\sigma$. Left: Noisy surface of unit diameter taken as $\mathcal{Z}$. Second left to right: Image $\Pi_\sigma(\mathcal{Z})$ under learned projections for different values of $\sigma \in \{0.01, 0.02, 0.04\}$. A larger choice of $\sigma$ leads to a projection onto a smoother surface.

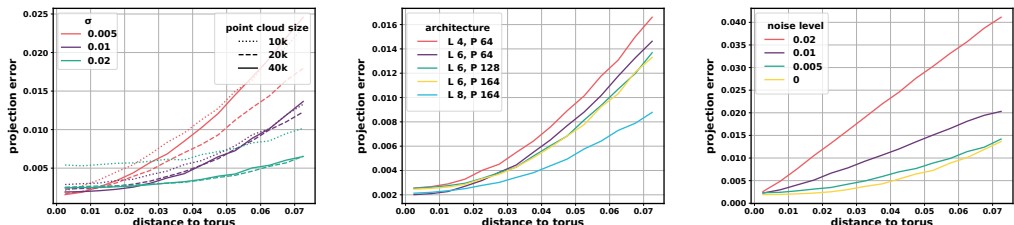

Figure 10: Error evaluation of learned torus projections versus distance to torus surface (which has unit outer radius). Left: Error for different sample sizes of torus and different values of $\sigma$; smaller $\sigma$ leads to a higher accuracy close to the surface and larger errors at certain distance. Middle: Error for different number of layers (L) and parameters per layer (P). Right: Error for training the projection on a torus surface with added Gaussian noise.

### A.4 DETAILS: DISCRETE SHELLS / ISOMETRIC AUTOENCODER

We provide additional details on how we learn a latent manifold representing a submanifold of the shape space of discrete shells for the results given in section 5.1.

**Data.** In this example, we use the SCAPE dataset (Anguelov et al., 2005) consisting of 71 immersions of a triangle mesh with 12500 vertices. To speed up the numerical algorithms used to create the samples for our autoencoder training, we reduced the resolution of the mesh using an iterative edge collapse approach to 1250 vertices. For visualization, we prolongated our results from the coarse to the fine mesh using a representation of the fine mesh vertices in terms of intrinsic positions and normal displacement with respect to the coarse mesh.

**Constructing the submanifold $\mathcal{M}$.** We begin by performing Principal Geodesic Analysis (PGA; Fletcher et al. (2004)) on the input dataset. This involves three steps: (i) computing the Riemannian center of mass (the mean shape), (ii) mapping each input shape to the tangent space at the mean via the discrete Riemannian logarithm, and (iii) applying Principal Component Analysis (PCA) to the resulting tangent vectors to obtain a low-dimensional linear subspace. The nonlinear submanifold is then recovered by applying the exponential map to this linear subspace.

All computations in this stage follow established numerical algorithms: For the Riemannian center of mass, the logarithms, and the exponential map, we use the methods introduced by Heeren et al. (2014) with time resolution $K = 8$. For the tangent PCA, we use the representation using edge lengths and dihedral angles as described by Sassen et al. (2020). We used the first two components for the example in fig. 1 and the first ten components for the example in figs. 4 and 11. To this end, we employed their publicly available C++ implementation (Heeren & Sassen, 2020).

**Learning the submanifold.** The training objective combines the reconstruction loss with the isometry loss introduced by Braunsmann et al. (2021) (see below). To generate training samples $\mathcal{X}$, we proceed as follows: First, we uniformly sample points within a hyperball of the linear tangent subspace. The size of the hyperball was chosen based on the norms of the projections of the input Riemannian logarithms onto the subspace. Second, for each such point, we sample a nearby point using a normal distribution with small variance centered on the first point. Finally, we apply the

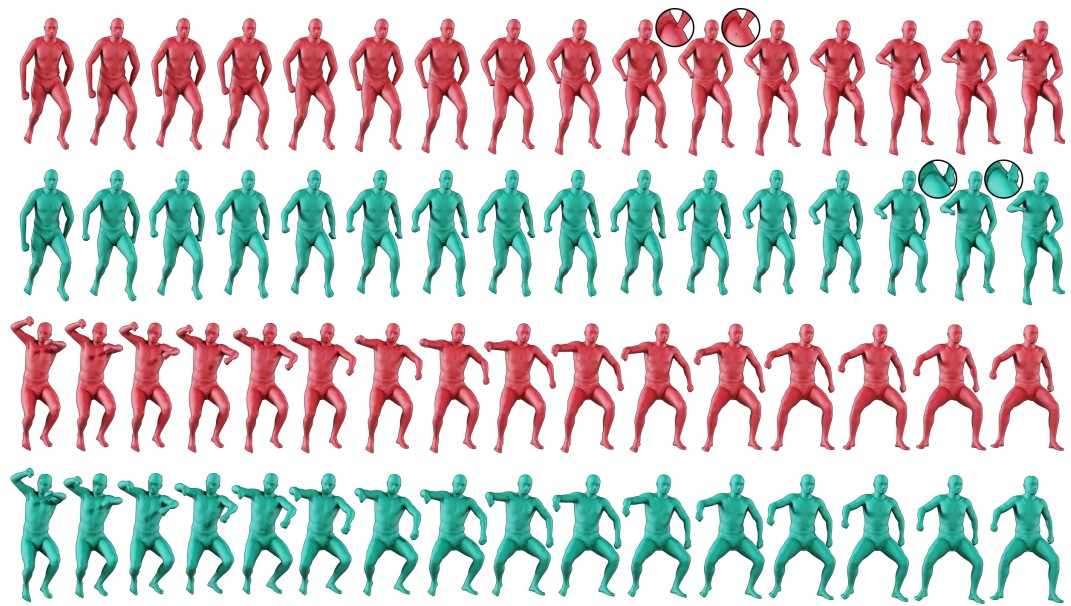

Figure 11: Two comparisons between linear interpolation (red) and geodesic interpolation using a learned projection on the embedded submanifold of discrete shells (green).

discrete exponential map to the two points to obtain a pair of points on the submanifold. We discard any pairs where at least one shape exhibits self-intersections to preserve physical plausibility. For the isometry loss, we compute the distance between the two shapes in a pair by computing discrete geodesics and taking their length. For the example in fig. 1, we drew $10000$ pairs this way, and, for the example in figs. 4 and 11, we drew $100000$ pairs. All these computations were performed with the same setup as described above.

The autoencoder architecture is a fully connected network with ELU activations. The encoder and decoder each have five layers, with the encoder reducing the input dimension from $3750$ (three times the number of vertices) to a latent dimension of $24$ and the decoder expanding it back accordingly.

**Projection learning.** We learn the projection on the embedded samples using $\sigma = 0.05$, six fully connected layers with $128$ intermediate dimensions, and ELU activations.

In fig. 11, we provide additional comparisons between linear interpolation in the latent space and geodesic interpolation using our learned implicit representation.

## A.5 DETAILS: MOTION CAPTURE DATA / SPHERICAL VARIATIONAL AUTOENCODER

We provide additional details for the motion capture experiment described in section 5.2.

**Data.** We use the sequences of subject 86 trial 1-6 from the CMU Graphics Lab Motion Capture Database (CMU Graphics Lab). These are approximately 52000 frames. We define a pose as an element of $\mathrm{SO}(3)^m$ as described in the main text and transform the data to this representation using an AMC parser (Zhou). We take $80\,\%$ as training data and $20\,\%$ for testing.

**SVAE network.** We use the pythae framework (Chadebec et al., 2022) for implementing an *SVAE* network with a decoder that decodes to vMF distributions without fixed variances. We use $l = 10$ latent dimensions. We optimize with AdamW (Loshchilov & Hutter, 2017) using a batch size of 100, an initial learning rate of $10^{-3}$, and an adaptive learning rate scheduler with a patience of 10 and reduce by a factor of 0.05. We use two fully connected layers to learn the embedding with dimensions (90, 30, 10) and a separate second layer (30, 1) for the variance. For the decoder, we

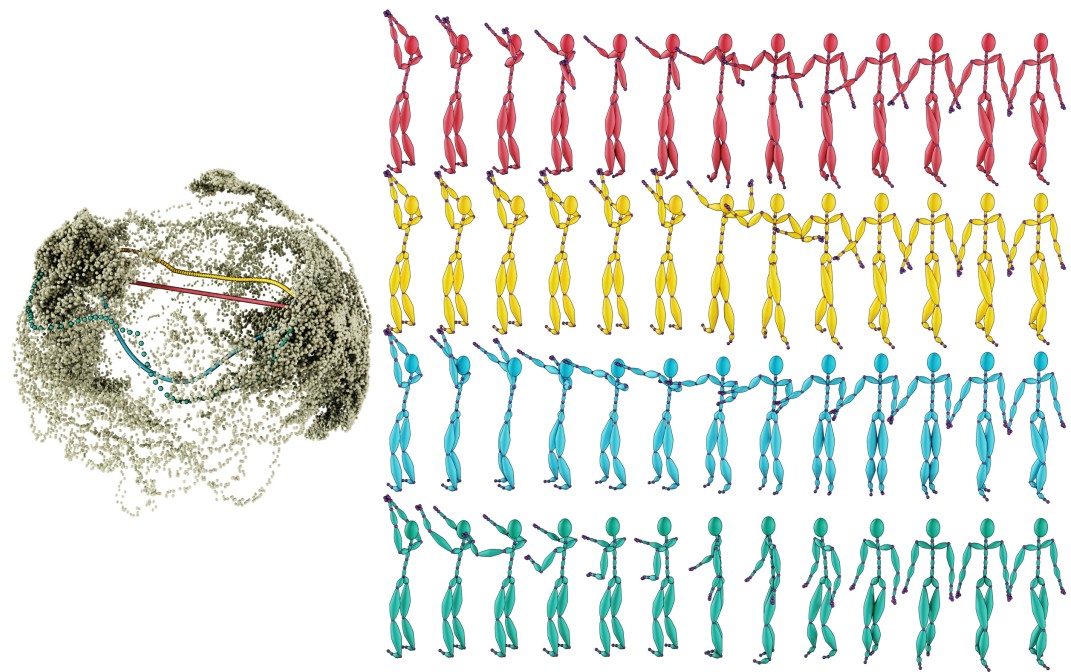

Figure 12: Same as fig. 5 for a different pair of endpoints. Left: Visualization of sample points in latent space (projected from $\mathbb{R}^{10}$ into $\mathbb{R}^3$) and computed paths between two points via linear interpolation (red), geodesic interpolation with $\mathcal{W}_E$ (yellow), unconstrained interpolation with $\mathcal{W}_{\mathcal{M}}$ (blue), and geodesic interpolation with $\mathcal{W}_{\mathcal{M}}$ (green). Right: Corresponding decoded sequences.

have dimension $(10, 30, 90)$ followed by one layer $(90, 90)$ for the decoded mean and one layer $(90, 30)$ for the decoded variances.

**Projection learning.** We train the projection on the embedded samples by embedding the full set of training data and sampling one point per resulting vMF distribution. We use a fully connected network with layers $(10, 64, 64, 64, 10)$ and choose $\sigma = 0.05$.

In fig. 12, we provide an additional example of geodesic interpolation. Moreover, as further variant, in this figure we also show a path computed using the pullback metric $\mathcal{W}_{\mathcal{M}}$ but without using the implicit representation $\zeta_\sigma$ as constraint.

A.6    DETAILS: IMAGE DATA / LOW BENDING, LOW DISTORTION AUTOENCODER

We provide additional details for the experiment described in section 5.3.

**Data.** We use the code provided by Braunsmann et al. (2024) to generate 30000 colored images with resolution $128 \times 128$ showing random rotations of the cow model.

**Low bending and low distortion autoencoder.** Each image $x$ corresponds to a specific rotation $r_x \in \mathrm{SO}(3)$. Hence, a distance between the images can be defined as $\mathrm{dist}_{\mathcal{M}}(x, y) = \arccos(r_x \cdot r_y)$ and geodesic averages $\mathrm{av}_{\mathcal{M}}(x, y)$ as renderings of the object with the mean rotation between $r_x$ and $r_y$. The autoencoder is trained with tuples of nearby points $(x, y) \in \mathcal{X}_\epsilon$, where $\mathcal{X}_\epsilon \subset \{(x, y) \in \mathcal{M} \times \mathcal{M} \mid \mathrm{dist}_{\mathcal{M}}(x, y) \leq \epsilon\}$. The regularization loss $\mathcal{J}_{\mathrm{reg}}$ for the encoder is given by

$$\mathcal{J}_{\mathrm{reg}}(\phi) = \frac{1}{|\mathcal{X}_\epsilon|} \sum_{x,y \in \mathcal{X}_\epsilon} \gamma\left(\frac{|\phi(x) - \phi(y)|}{\mathrm{dist}_{\mathcal{M}}(x, y)}\right) + \lambda \frac{|\phi(\mathrm{av}_{\mathcal{M}}(x, y)) - \mathrm{av}_{\mathbb{R}^l}(\phi(x), \phi(y))|^2}{\mathrm{dist}_{\mathcal{M}}(x, y)^4},$$

where $\gamma(s) = |s|^2 + |s|^{-2} - 2$, $\mathrm{av}_{\mathbb{R}^l}(a, b) = (a+b)/2$ denotes the linear average, and $\lambda > 0$. The first term promotes an isometric embedding, encouraging $|\phi(x) - \phi(y)| = \mathrm{dist}_{\mathcal{M}}(x, y)$. The second term

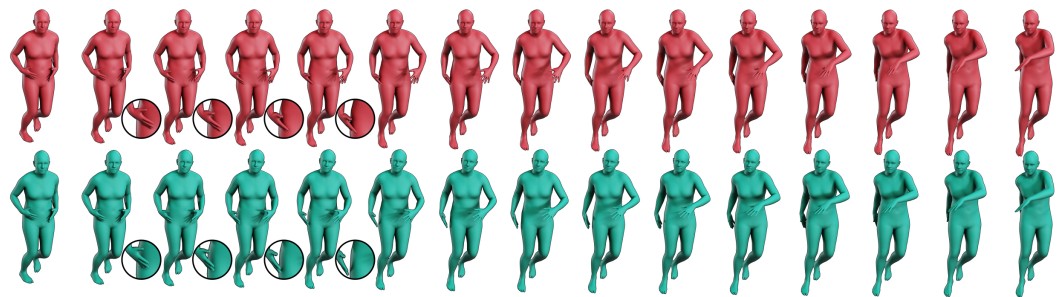

Figure 13: Linear interpolation in a quaternion representation of the SMPL body model (red) and geodesic interpolation using the neural distance function Pose-NDF to a manifold of plausible poses (green).

penalizes the deviation between the embedding of the manifold average and the Euclidean average of the embedded points, favoring a flat embedding. For details, we refer to Braunsmann et al. (2024).

We use the publicly available pretrained model for flatness weight $\lambda = 10$ and $l = 16$ latent dimensions.

**Projection learning.**     We learn the projection on the embedded samples using $\sigma = 0.005$, eight fully connected layers with 128 intermediate dimensions, and ELU activation functions.

## A.7    DETAILS: POSE-NDF

Figure 13 shows an additional example using the neural distance function Pose-NDF (Tiwari et al., 2022) as manifold representation, see section 5.4. We compare linear interpolation in the quaternion representation of the SMPL body model used in Tiwari et al. (2022) and geodesic interpolation on the manifold corresponding to the approximate zero-level set.

