# OpenReview forum: "Geodesic calculus on latent spaces"
_ICLR.cc/2026/Conference — ICLR 2026 Conference Withdrawn Submission_

### Official Review · Reviewer_Dcdf · 2025-10-24

**Soundness:** 3
**Presentation:** 3
**Contribution:** 2
**Rating:** 4
**Confidence:** 4

**Summary:**

This paper introduces a new perspective on latent spaces as Riemannian manifold and offers the associated tools to do latent interpolation and extrapolation. This approach assumes that latent spaces are noisy representations of latent Riemannian manifolds defined by an immersion obtained with an autoencoder. It describes how to learn the immersion by learning an approximate of the projections on the latent manifold. The resulting interpolation and extrapolation are qualitatively better than their linear (Euclidean) counterparts.

The description of latent spaces as immersed Riemannian Manifold has shown to be an interesting idea.
Learning of the projector associated to the immersion is the key new idea.
The authors propose an augmented Lagragian approach to compute the discrete geodesic associated with the approximate distance function derived from a Riemannian metric
The use of the Lagrangian scheme is also used to provide the Exp map of the manifold.

**Strengths:**

The paper is well written and showcases multiple use cases on different types of data.
It also demonstrates interesting connection between variational calculus and Riemannian geometry with theoretical groundings.
The immersion approach alleviates the issue of data manifold support often encountered with other methods. Thus, it does not require ad-hoc calibration like in pullback methods or explicit support using centroids.
The projection operator is claimed to be by design robust to imprecise representation, which ensures the interpolants will be correctly decoded.
Clear and well supported geodesic and Exp map computations.

**Weaknesses:**

Albeit the paper showcases some interesting inter/extra-polated trajectories, it fails to quantitatively demonstrate why these should be favored.

More generally the paper, whilst proposing interesting insight, fails to compare itself with other geodesic solvers, see GEORCE (https://arxiv.org/abs/2505.05961v1) or a graph based approach (for instance: https://arxiv.org/pdf/2407.11244).
For example, Fig.8 suggests that geodesic interpolation avoids self-intersection. How much is this the case? It could be interesting to count these self-intersections under different framework to see how the proposed method compares.

The projector operator relies on a well-tuned bandwidth, the effect of which is investigated in A.3. Yet these expected insights fail to transfer to actual impact on the geodesic quality. Quantitatively assessing the geodesics quality with regards to the performance of the projector could help in finding rule of thumb to set $\sigma$.

The paper could benefit from a concise complexity analysis. The minimization of Eq.9 can be complex and expansive. For example, even using the approximate distance of Eq.5 yields infeasible computation as it will require computing the Jacobian of a (possibly large) decoder K times at each iteration. This issue is even worse for the Exp map as it requires second order minimization. To demonstrate the paper claims that it offers practical geometric solutions, it could be interesting to report clock-time/memory requirements/scaling with time resolution and latent dimension.

Also, the code is not submitted with the supplemental albeit it is mentioned as one of the core contributions of the paper.

**Questions:**

For the extrapolations, it is clear that under Euclidean geometry (no projection) the trajectory will diverge, but is it still the case using Euclidean geometry whilst enforcing the immersion constraint? Studying this on known non-isometric networks could show the separate effect of the immersion approach from the choice of the metric.

In section 5.4, it reads that even when having only access to a signed function it is possible to compute geodesic. Yet if $d$ is not smooth or differentiable, how is Eq.9 actually computed? I would assume the minimization of the Lagrangian uses 1st order methods and that the constraint $ \zeta(z)=0$ forces to work on the domain where $d$ is not differentiable.

What happens to the assumption $\zeta \approx \zeta_\sigma$ when the inference points are far from the data manifold?

How is the Exp map computed when we only have access to $z_0$ and $\dot{z_0}$?

The whole approach is not applicable when the only information we have on the manifold is the actual metric tensor, like in this paper . In which case, eg Eq.4, to compute the geodesic we need to already be able to compute geodesic distances. This chicken and egg problem should at least be mentioned.


Minor comment :
In Eq.8 , I think it should be $\mathcal{E}^K$ and not $\mathcal{E}$.

---

### Official Review · Reviewer_zd4H · 2025-10-24

**Soundness:** 3
**Presentation:** 2
**Contribution:** 2
**Rating:** 4
**Confidence:** 3

**Summary:**

The latent spaces of autoencoders have interesting geometric properties. The authors propose the scheme for computing the exponential maps and logarithmic maps in discrete time. The latent spaces of autoencoders are modeled using a submanifold of some unknown ambient manifold, where the projection operator onto the submanifold is learned. Experiments validate the applicability of the proposed method for learning the geometric properties of data and for performing interpolations.

**Strengths:**

1. The proposed methods are motivated based on theory.
2. The empirical evidence validates the applicability of the methods.

**Weaknesses:**

1. In my view the modeling method is not that well motivated. See Questions below.
2. The computation methods for the exponential maps and logarithmic maps are also not that well-motivated. See Questions below.

**Questions:**

For modeling, I assume the goal is to learn an approximate geometry based on the data distribution. However, the authors propose to use the denoising objective to learn a submanifold inside the latent space, while I could not find strong motivation for this design choice. Furthermore, it is unclear to me why we should prefer an implicit representation instead of an explicit one which gives the chart.

As a general comment, numerical methods for solving the exponential maps and logarithmic maps have been proposed before, see e.g. https://github.com/MachineLearningLifeScience/stochman.

One can in principle compute the exponential maps by numerically solving the geodesic equation, which should in principle recover the continuous limit of $K \rightarrow \infty$. It would be great if further justifications can be provided on why should the proposed discrete approach be preferred instead of solving the geodesic equation; e.g. is it the case that we do not have access to the intrinsic coordinates?

Previous works already compute the logarithmic maps on embedded manifolds by parameterizing splines and optimize its parameters; see e.g. the GitHub repository. It would be great if the authors can comment on the connections.

---

### Official Review · Reviewer_jyJr · 2025-10-30

**Soundness:** 1
**Presentation:** 2
**Contribution:** 1
**Rating:** 2
**Confidence:** 4

**Summary:**

This paper proposes to characterize latent manifold as implicit manifolds via a learned approximate projection, which is then used to compute discretized geodesics with an augmented Lagrangian method. The obtained geodesics follow the manifold of interest, in contrast to Euclidean geodesic (a.k.a straight lines).

**Strengths:**

The paper targets a relevant problem, i.e., computing geodesics and considering the geometry of latent spaces. I appreciate the various experiments presented to illustrate the proposed method.

**Weaknesses:**

My main concern with the paper is the weakness of its contributions compared to previous works, coupled with a lack of experimental comparisons.

Various related works have considered the geometry of the data manifolds, typically by pulling back the Riemannian metric of the ambient space through a decoder, represented either via a Gaussian process (Tosi et al), or via a neural network (Chen et al, Arvanitidis et al). The obtained Riemannian metric entirely defines the geometry of the latent manifold and can be used to compute geodesic that stays on the data manifold. In contrast, the paper learns an approximate projection onto the latent manifold. While this indeed allows the projection of latent points onto the manifold, the paper still requires the latent space to be endowed with a Riemannian metric, i.e., the projection itself is not enough to define the geometry of the manifold and compute associated geodesics. In this paper, the Riemannian metric is either chosen empirically as flat (Sec. 5.1, 5.3), or obtained as the pullback metric similar to previous work. The former requires additional assumption on the latent space and/or the AE which do not always apply, while the latter boils down to pulling back the metric as in previous work thus resulting in a two-step approach (instead of the single step required in previous works such as Arvanitidis et al). Thus, the advantages compared to previous works are unclear.

Moreover, the requirements or benefits of computing discrete rather than continuous geodesics in the considered setup are unclear to me. As the metric of the latent space is known, geodesics could be computed as in previous works, i.e., by solving an ODE, approximating the geodesic as a spline (e.g., Tosi et al, Arvanitidis et al), or in a discrete fashion via a Dijkstra algorithm as in (Beik-Mohammadi et al, 2023, Sec. 4.4.3). However, the paper does not compare to these alternatives approaches neither theoretically nor experimentally. Such comparisons are crucially missing, and the contributions of the paper, as well as the performance of the proposed geodesic computation approach compared to previous works, remain unclear. In addition, the aforementioned related works only require a Riemannian metric on the latent space to compute geodesics, while the proposed approach requires both the Riemannian metric and the learned projection.


References:
- Beik-Mohammadi, H., Hauberg, S., Arvanitidis, G., Neumann, G. & Rozo, L. Reactive motion generation on learned Riemannian manifolds. Int. J. Robot. Res. 42, 729–754 (2023).

**Questions:**

- What are the advantages of learning an approximate projection?
- How does the proposed geodesic computation method compare to the state of the art (precision, computational time, etc)?

---

### Official Review · Reviewer_CuS3 · 2025-10-31

**Soundness:** 2
**Presentation:** 2
**Contribution:** 2
**Rating:** 4
**Confidence:** 4

**Summary:**

The authors consider an encoder-decoder neural network with a latent space of dimensionality l. Within this latent space, they identify a submanifold and investigate how to compute geodesic paths on it. The metric is defined through the discrete path energy given in Eq. (2), while the projection onto the manifold is learned following the approach of Alain & Bengio (2014). Experimental results demonstrate that the proposed method produces more realistic and smoother interpolation between data samples compared to linear interpolation in latent space.

**Strengths:**

The paper presents a well-engineered approach that combines established algorithms to simultaneously identify the manifold structure and compute geodesic paths within it The resulting embeddings produce realistic image transformations in human motion data that evolve smoothly over time. The algorithms used are standard and appear highly reproducible, making the proposed method suitable for applications in computer graphics and related domains.

**Weaknesses:**

The justification for using the objective functions and the denoising autoencoder framework proposed by Alain & Bengio (2014) should be elaborated further. There should be many other combination of methods that can produce similar results.

Can the authors provide additional experimental results on applications beyond human motion data? How robust is the proposed method for interpolating other types of time series data? How does the algorithm perform in tasks such as video frame interpolation or medical image reconstruction?

**Questions:**

The authors employ denoising manifold projection proposed by Alain & Bengio (2014), which was originally designed for use on the data space rather than on embedded representations. The rationale for applying this method in the latent space should be clearly justified and theoretically supported.

---

### Note · Authors · 2025-11-19

**Comment:**

We appreciate the reviewers' time and feedback on our submission. After careful consideration of the reviews, we have decided to withdraw the paper. We plan to comprehensively address the raised questions and concerns and will use them to thoroughly revise and improve our work for a future submission.
Here, we would like to briefly comment on some aspects of the reviews.

In general, methods to compute geodesics on latent manifolds need both a metric on the manifold and a computationally accessible representation of the manifold as a set in the ambient latent space, in contrast to computations carried out in the unrestricted full latent space.
Our focus is on the design of an easy to use tool for this representation that is flexible to be integrated in an existing autoencoder framework without the need to modify the encoder or decoder.

Regarding our approach to compute geodesics, we would like to clarify that our "discrete geodesics" are a numerical discretization of the continuous geodesics as solutions of a variational problem on the latent manifold.
The discrete scheme provides a provably convergent way to approximate the continuous geodesics as minimizers of the path energy for given end points.
We will extend the discussion and comparison with other methods.

We model the latent manifold as an implicit manifold because it provides a unified description of the geometry and scales to higher-dimensional or topologically nontrivial settings. In contrast, methods that rely on explicit parametrizations and local coordinates require multiple charts and mappings for the change of charts, which can be demanding to handle. This is also why the mentioned geodesics solvers are not directly applicable in our setting.

Using the denoising objective as a tool to learn the implicit representation is a choice that allows for an easy to use method and works for a variety of applications.

**Withdrawal Confirmation:**

I have read and agree with the venue's withdrawal policy on behalf of myself and my co-authors.